# Therapeutic, Clinicopathological, and Molecular Correlates of *PRKACA* Expression in Gastrointestinal Cancers

**DOI:** 10.3390/ph17101263

**Published:** 2024-09-25

**Authors:** Ayoub Al Othaim, Glowi Alasiri, Abdulaziz Alfahed

**Affiliations:** 1Department of Medical Laboratories, College of Applied Medical Sciences, Majmaah University, Al-Majmaah 11952, Saudi Arabia; ay.alothaim@mu.edu.sa; 2Department of Biochemistry, College of Medicine, Imam Mohammad Ibn Saud Islamic University (IMSIU), Riyadh 13317, Saudi Arabia; gaalasiri@imamu.edu.sa; 3Department of Medical Laboratory, College of Applied Medical Sciences, Prince Sattam bin Abdulaziz University, Alkharj 11942, Saudi Arabia

**Keywords:** gastric cancer, colorectal cancer, *PRKACA* expression, clinicopathological features, gene set enrichment analysis, drug target enrichment analysis, gene ontology enrichment analysis, *PRKACA* deregulation

## Abstract

Background/Objectives: *PRKACA* alterations have clear diagnostic and biological roles in the fibrolamellar variant of hepatocellular carcinoma and a potential predictive role in that cancer type. However, the roles of *PRKACA* have not been comprehensively examined in gastric and colorectal cancers (GC and CRC). This study, therefore, sought to investigate the roles of *PRKACA* expression in GC and CRC. Methods: The clinico-genomic data of 441 GC and 629 CRC cases were analyzed for therapeutic, clinicopathological, and biological correlates using appropriate bioinformatics and statistical tools. Furthermore, the deregulation of *PRKACA* expression in GC and CRC was investigated using correlative and regression analyses. Results: The results showed that *PRKACA* expression subsets were enriched for gene targets of chemotherapeutics, tyrosine kinase, and β-adrenergic inhibitors. Moreover, high *PRKACA* expression was associated with adverse clinicopathological and genomic features of GC and CRC. Gene Ontology Enrichment Analysis also showed that *PRKACA*-high subsets of the GI cancers were enriched for the biological and molecular functions that are associated with cell motility, invasion, and metastasis but not cell proliferation. Finally, multiple regression analyses identified multiple methylation loci, transcription factors, miRNA species, and *PRKACA* copy number changes that deregulated *PRKACA* expression in GC and CRC. Conclusions: This study has identified potential predictive and clinicopathological roles for *PRKACA* expression in GI cancers and has added to the growing body of knowledge on the deregulation of *PRKACA* in cancer.

## 1. Introduction

Gastrointestinal (GI) cancers remain important public health concerns despite the huge number of resources that have been expended in research efforts to decipher the clinical, molecular, and biological characteristics of these cancers [1,2,3]. Gastric and colorectal cancers (GC and CRC), as a group, comprise the commonest malignancies worldwide ahead of the female breast, lung, and prostate cancers, accounting for over 3.0 million (15.33% of all malignancies) new cases [4,5] and constitute the 2nd commonest cause of cancer deaths with 1,703,966 (16.76% of all malignancies) new deaths [4,5]. An improved understanding of the tumor biology of GI cancers will inform innovative and enhanced strategies for cancer management [6,7,8]. Hence, new research that illuminates GI carcinogenesis and progression is warranted.

This study investigated the potential therapeutic significance of *PRKACA* expression in GC and CRC, as well as its clinicopathological and molecular correlates. *PRKACA* encodes the catalytic subunit-α of PKA, a kinase enzyme that functions downstream of the β-receptors in the β-adrenergic signaling pathway [9,10,11,12]. PKA phosphorylates multiple downstream targets in the β-adrenergic signaling pathway, including transcription factors *ATF*, *CREB*, *GATA1*, *STAT3*, Src, and *BARK* [9,11,12]. However, the roles of *PRKACA* in cancer are incompletely understood, inasmuch as the cyclic adenosine monophosphate (cAMP)-PKA signaling has been shown to possess both tumor-suppressive and oncogenic roles in different tumor types and contexts [12]. A preclinical study demonstrated that crotonylation of *PRKACA* enhances the activity of PKA and thereby promotes colorectal cancer development via the *PKA*-*FAK*-*AKT* pathway, whereas *PRKACA* is secreted by prostate cancer cells [11]. In clinical cancer, *PRKACA* participates in a fusion event with *DNAJB1* in up to a hundred percent of fibrolamellar hepatocellular carcinoma and has therefore been identified as a suitable diagnostic marker for and a driver of oncogenic activities in that cancer [13,14,15]. In addition, *PRKACA* was found to be upregulated in the serum and tumor of gastric and colorectal cancer patients but did not show any clinical correlates in a study that was markedly limited by sample sizes [16]. The therapeutic significance of *PRKACA* expression in cancer had previously been suggested by a breast cancer study, which showed that *PRKACA* mediates resistance to anti-HER2 therapy in breast cancer cell lines via inactivation of BAD and Bcl2-associated death promoter [17]. Indeed, elevated *PRKACA* expression was found in trastuzumab-resistant breast cancer patients, evidence that *PRKACA* is activated in trastuzumab-resistant breast cancer cases [16]. With respect to therapy, *PRKACA* could be a target for the treatment of cancer. For example, it has been suggested that the PAK signaling and *PRKACA* activities could be inhibited upstream using β2 receptor blockade with propranolol as a strategy for the treatment of invasive cancer [9]. In addition, Schalm et al. and Bauer et al. [14,18] demonstrated that *PRKACA*—in the *DNAIJ-PRKACA* fusion context—is a viable target for the treatment of the *PRKACA*-dependent fibrolamellar variant of hepatocellular carcinoma. However, the biological and therapeutic significances of *PRKACA* expression in clinical GC and CRC have not been thoroughly interrogated. Therefore, the aims of this study include (i) to interrogate the potential therapeutic significance of *PRKACA* in clinical GC and CRC, (ii) to investigate the clinicopathological and genomic correlates of *PRKACA* expression in GC and CRC, (iii) to investigate the biological significance of *PRKACA* expression in GC and CRC, (iv) to investigate the mechanisms of *PRKACA* deregulation in GI cancers.

## 2. Results

### 2.1. Enrichment of Gene Targets of Multiple Drug Agents in PRKACA-Low and -High Cancers

#### 2.1.1. Gastric Cancer Cohort

Normalized *PRKACA* expression values were dichotomized using the mean *PRKACA* values for each cohort. There were 206 *PRKACA*-low (50.0%) and 206 *PRKACA*-high (50.0%) GC cases. GSEA followed by drug set annotations on Enrichr (https://maayanlab.cloud/Enrichr/enrich, accessed on 25 July 2024) revealed enrichment of genes that are targets of multiple drugs, including tyrosine kinase inhibitors, β-blockers, and conventional chemotherapeutic agents, among other drug classes, in both the high and low *PRKACA* subsets, and an adjusted *p* value of ≤0.05 and false discovery rate (FDR) of ≤0.05 (Figure 1). Gene targets enriched in the *PRKACA*-high subset include those for conventional chemotherapeutic agents such as paclitaxel, irinotecan, camptothecin, ifosfamide, and methotrexate; tyrosine kinase inhibitors sorafenib, dasatinib, axitinib, gefitinib; β-blockers such as atenolol, and metoprolol; and calcium channel blockers, such as dexverapamil, verapamil and diltiazem (Appendix A Drug-targets_PRKACA_high_GC). *PRKACA*-low GC subset, on the other hand, shows significant enrichment of targets that are downregulated following treatment with chemotherapeutic agents such as irinotecan, camptothecin, and daunorubicin (Figure 1). Furthermore, the *PRKACA*-low subset was enriched for gene targets associated with tyrosine kinase inhibitors such as momelotinib, ponatinib, sunitinib, dasatinib, sorafenib, vandetanib, among many others (Appendix A Drug-targets_PRKACA_low_GC). However, it was observed that one phenotype (*PRKACA*-high, for example) might be associated with both upregulation and downregulation of the gene set of the same drug, depending on the library that was interrogated.

#### 2.1.2. Colorectal Cancer Cohort

Normalized *PRKACA* expressions were dichotomized into *PRKACA*-low (N = 268, 50%) and *PRKACA*-high subsets (N = 268, 50%). Significant enrichment of gene targets of multiple drugs was observed in both *PRKACA* subsets in the CRC cohort. Similar to the GC cohort, *PRKACA* expression in the CRC cohort exhibited enrichment of genes that are associated with β-blockers, conventional chemotherapeutics, and tyrosine kinase inhibitors (Figure 1). Furthermore, as was observed in the GC cohort, the pattern of upregulation or downregulation of drug targets differs with the gene sets or libraries interrogated. For example, in the *PRKACA*-high subset, irinotecan gene sets showed both downregulated or upregulated targets depending on the library that was interrogated. This finding may have resulted from the differences in the biology of the cancer cell lines (e.g., breast cancer (MCF7) vs. leukemia (HL60) cell lines) that were utilized to generate the gene set libraries (Appendix A Drug-targets_PRKACA_high_CRC and Appendix A Drug-targets_PRKACA_low_CRC).

### 2.2. Clinicopathological and Molecular Features of PRKACA Expression

#### 2.2.1. Gastric Cancer Cohort

The clinicopathological and molecular correlates of *PRKACA* expression were interrogated using the Chi-square test. The results showed that in the GC cohort, high *PRKACA* expression was significantly associated with the 60-years-and-below age group, higher nodal stage, and late disease stage but not with gender, histological tumor type, pathological tumor stage, or distant metastasis (Table 1). *PRKACA* expression did not show any association with a prognosis on Kaplan-Meier analysis (Log Rank X2 = 1.832, *p* = 0.176). Furthermore, high *PRKACA* expression was associated with the genome-stable and Epstein-Barr Virus subtypes of GC and with tumors having low aneuploidy and high global methylation scores (Table 2).

#### 2.2.2. Colorectal Cancer Cohort

In the CRC cohort, high *PRKACA* expression was associated with younger age groups (particularly the 60-years-and-below age group), left-sided colon cancer, rectal cancer, and nodal stage, but not with gender, race/ethnicity, pathological tumor stage, metastasis, TNM stage, lymphovascular and vascular invasion, and histological type (Table 1). As with the GC cohort, *PRKACA* expression did not show any association with overall or progression-free survival (OS: Log Rank X^2^ = 0.039, *p* = 0.844; DFS: Log Rank X^2^ = 0.054, *p* = 0.816). Moreover, high *PRKACA* expression was associated with the chromosomal instability and microsatellite stable (MSS) subtypes of CRC and with tumors having low mutation count, low MSI Sensor score, and high Fraction Genome Altered and Aneuploidy scores, but not with MSI MANTIS score and tumor mutational burden (TMB) (Table 2).

### 2.3. Biological Significance of PRKACA Expression

#### 2.3.1. Gastric Cancer Cohort

GSEA showed significant differential enrichment of HALLMARK_MYOGENESIS, HALLMARK_APICAL_JUNCTION, HALLMARK_EPITHELIAL_MESENCHYMAL_TRANSITION, and HALLMARK_COAGULATION gene sets in the *PRKACA*-high subset of GC cohort (Appendix A GC_GSEA.PRKACA_high). GO Enrichment Analysis showed that the enriched genes identified have a biological function in Cell-Matrix Adhesion (GO:0007160), Regulation of Cell Migration (GO:0030334), Actomyosin Structure Organization (GO:0031032), Contractile Actin Filament Bundle Assembly (GO:0030038), Regulation of Cell Motility (GO:2000145), etc., functions which are involved in cell motility, migration and metastasis (Figure 2 and Appendix A GOEA_GC_PRKACA_high). On the other hand, the *PRKACA*-low GC subset showed differential enrichment of the HALLMARK_G2M_CHECKPOINT, HALLMARK_MTORC1_SIGNALING, HALLMARK_PROTEIN_SECRETION and HALLMARK_E2F_TARGETS gene sets (Appendix A GC_GSEA.PRKACA_low). GO Enrichment Analysis identified the core enrichment genes in these gene sets to have such biological functions as Mitotic Cell Cycle Phase Transition (GO:0044772), Mitotic Spindle Organization (GO:0007052), DNA-templated DNA Replication (GO:0006261), etc., all of which are associated with cell replication, proliferation and tumor growth (Figure 2 and Appendix A GOEA_GC_PRKACA_low). The results of the analyses suggest that *PRKACA* may be a pro-migration/invasion and anti-proliferation gene in gastric carcinogenesis and gastric cancer progression.

#### 2.3.2. Colorectal Cancer Cohort

GSEA showed differential enrichment of the HALLMARK_MYOGENESIS, HALLMARK_APICAL_JUNCTION, HALLMARK_KRAS_SIGNALING_DN, HALLMARK_COAGULATION, HALLMARK_ANGIOGENESIS, HALLMARK_EPITHELIAL_MESENCHYMAL_TRANSITION, HALLMARK_APICAL_SURFACE, HALLMARK_HEDGEHOG_SIGNALING, HALLMARK_NOTCH_SIGNALING, and HALLMARK_IL6_JAK_STAT3_SIGNALING gene sets in the PRKACA-positive CRC cohort (Appendix A CRC_GSEA.PRKACA_high). GO Enrichment Analysis showed that the enriched genes identified in the GSEA have biological functions, which included Cell-Matrix Adhesion (GO:0007160), Actomyosin Structure Organization (GO:0031032), Positive Regulation of Cell Migration (GO:0030335), Positive Regulation of Cell Motility (GO:2000147), etc. These biological functions are associated with cell motility, invasion, and metastasis, as noted above in the GC cohort (Figure 3 and Appendix A GOEA_CRC_PRKACA_high).

The *PRKACA*-low CRC subset did not show differential enrichment of any of the HALLMARK gene sets associated with cell proliferation.

### 2.4. Differential Core Enrichment Gene Sets between GC and CRC PRKACA-High Subsets

A total of 281 genes comprising core enrichment components from the five Hallmark gene sets were found for the GC *PRKACA*-high subset, while 503 genes were enriched in the CRC *PRKACA*-high subset (Appendix A Core.Enrichment_GeneSets). A comparison of the core enrichment gene sets in the GC and CRC *PRKACA*-high subsets showed that only two enriched genes, *EGFR* and *LAMP2*, overlapped between the GC and CRC subsets, whereas the GO enrichment analysis found that the enriched biological processes, molecular functions, and cellular components subserved by the enriched genes were similar between the GI cancers. This is evidence that *PRKACA* or the PKA signaling may regulate diverse genes/genetic pathways in different cancer types to accomplish the same biological processes and molecular functions.

### 2.5. Deregulation of PRKACA Expression in Gastrointestinal Cancers

#### 2.5.1. Gastric Cancer Cohort

*PRKACA* deregulation was investigated using copy number, methylation, miRNA, and transcription factor expression. *PRKACA* copy number alterations were obtained from the copy number segment data of the TCGA GC cohorts using the segment mean threshold of 0.3 (for gain/amplification) and −0.3 (for deletion). Using this threshold, the study identified 31/412, 375/412, and 6/412 *PRKACA* deletions, neutral, and gain/amplification, respectively. There was a significant association between *PRKACA* copy number changes and expression (X^2^ = 23.609, *p* < 0.001; Figure 4). A total of 20 *PRKACA* promoter methylation loci and their beta values were retrieved from the gastric cancer methylation data. Bivariate correlation analysis demonstrated that 4/20 methylation loci were correlated with *PRKACA* expression (see Appendix A GC_PRKACA_Methylation_loci). Gene enrichment analysis with DEseq2 identified the top 40 miRNA species that were differentially enriched in the *PRKACA*-high and -low cases in the GC cohort. Bivariate analysis showed that the expression of 20/40 miRNA species showed correlations with *PRKACA* expression (see Appendix A GC_PRKACA-targeting miRNA). A total of 56 *PRKACA*-targeting transcription factors were retrieved from the transcription factor database TF2DNA_DB. Bivariate analysis showed that the expression of 27/56 transcription factors was correlated with *PRKACA* expression in the GC cohort (see Appendix A GC_PRKACA_TranscriptionFactors).

*PRKACA* copy number changes, methylation loci beta values, *PRKACA*-targeting transcription factors, and the differentially enriched miRNA values were incorporated into a multiple regression analysis. The results showed that multiple regulatory mechanisms independently predicted *PRKACA* expression in the GC cohort (Table 3, Figure 5).

#### 2.5.2. Colorectal Cancer Cohort

In the CRC cohort, there were 10/480, 455/480, and 15/480 *PRKACA* copy number deletions, neutral, and gain/amplification, respectively. Furthermore, there were significant associations between *PRKACA* copy number changes and expression (X^2^ = 5.039, *p =* 0.007; Figure 4). The bivariate analysis identified 11/20 methylation loci whose beta values exhibited a significant correlation with *PRKACA* expression in the CRC cohort (see Appendix A CRC_PRKACA_Methylation_loci). Only one methylation locus, cg17818798, exhibited a shared *PRKACA* methylation-expression correlation between the CRC and GC cohorts. The expression of 25/40 miRNA species identified in the miRNA gene enrichment analysis of the CRC cohort showed correlations with *PRKACA* expression on bivariate analysis (see Appendix A CRC_PRKACA-targeting miRNA). The expression of seven (7) miRNA species, including hsa-let-7c, hsa-mir-1-1, hsa-mir-1-2, hsa-mir-133b, hsa-mir-143, hsa-mir-7641-1, and hsa-mir-99a were found to be commonly correlated with *PRKACA* expression in the GC and CRC cohorts. Of the 56 *PRKACA*-targeting transcription factors retrieved from the TF2DNA_DB, the expression levels of 26 showed a significant correlation with *PRKACA* expression (see Appendix A CRC_PRKACA_TranscriptionFactors). The expression levels of 16/56 *PRKACA*-targeting transcription factors retrieved from the TF2DNA_DB were found to be commonly correlated with *PRKACA* expression in both the GC and the CRC cohorts. These included *HOXA1*, *ZNF451*, *TFEB*, *TSHZ3*, *ZNF644*, *ZNF557*, *DHX34*, *HEY2*, *ZNF235*, *KLF15*, *HAND2*, *E2F7*, *TGIF1*, *ZBTB7A*, *MAFK*, and *CHAMP1*.

Multiple linear regression analysis showed that a combination of regulatory mechanisms, including *PRKACA* copy number alterations, transcription factor expression levels, promoter methylation status, and miRNA expression levels, independently predicted *PRKACA* expression in the CRC cohort (Table 4; Figure 5). The methylation locus cg17818798 and *ZNF451* and *ZNF557* expression were the predictors of *PRKACA* expression common to both the GC and CRC cohorts.

## 3. Discussion

*PRKACA*, the α-catalytic subunit of PKA, is an essential component of PKA signaling that catalyzes the downstream targets of PKA [9,12]. This study demonstrated that *PRKACA* expression in GC and CRC displays enrichment of genes that are associated with multiple conventional chemotherapeutic agents, FDA-approved tyrosine kinase inhibitors, and β-adrenergic blockers. The study findings support the role of *PRKACA* as a potential biomarker for multiple tyrosine kinase inhibitors, chemotherapeutic drugs, and β-adrenergic blockers. The findings also lend credence to the Cole and Sood study [9], which suggested that the inhibition of β-adrenergic signaling could possibly be utilized for the treatment of invasive cancer. This is even more so with the demonstration that high *PRKACA* expression was associated with the tumor hallmark of invasion or migration, rather than proliferation, in the GC and CRC cohorts. Moreover, the enrichment of gene targets of multiple drugs in *PRKACA*-high cases highlights the concept of an integrated, multitargeted approach to cancer therapy [19], wherein a single biomarker could at once be predictive of response to multiple drugs with varying mechanisms of action, on the basis of the association of that biomarker with a global gene expression network that incorporates the gene targets for those drugs [20]. The advantages of such an integrated approach to cancer therapy would include the augmentation of the therapeutic options for the oncologist and the reduction of the risks of drug toxicity and resistance without the violation of the tenets of personalized medicine [21,22]. To reiterate, the pro-migration/invasion characteristics of *PRKACA,* as found in this study, support a proposal for targeting *PRKACA* directly [14,18] and/or the β-adrenergic pathway [9] for treating tumor invasion.

The tumor-promoting activities of the *PRKACA* and PKA signaling have been described for classic and fibrolamellar variants of hepatocellular carcinoma, and breast, prostate, ovarian, and non-small cell lung cancers, glioblastoma, astrocytoma, and leukemia [11,12,13,14,15]. The association of *PRKACA* expression with some adverse clinicopathological features of gastrointestinal cancers, such as advanced age, tumor site, tumor histotype, nodal stage, and TNM stage, are in keeping with the tumor-promoting activities in the above-mentioned studies. Furthermore, this study showed that *PRKACA* expression is associated with poor prognosis molecular features and subtypes of gastrointestinal cancers. To the best of our knowledge, these findings have not been previously elucidated in any gastrointestinal cancer cohorts.

Interestingly, GSEA and Gene Ontology Enrichment Analysis demonstrated that while *PRKACA*-high tumors exhibited differential enrichment of genes that regulate cell motility, migration, and metastasis in the GC and CRC cohorts, there was enrichment of genes that regulate cell proliferation and growth in the GC, but not the CRC, *PRKACA*-low subset. The differential enrichment of actin filament organization, actin polymerization, actomyosin structure, focal adhesion structure, and membrane organelle organization are in keeping with the study by McKenzie et al. [23], which found that the PKA signaling is locally and rapidly activated by mechanical stretch in an actomyosin contractility-dependent manner, thus establishing the activated PKA as an effector of cellular mechanotransduction and a regulator of mechanically guided cell migration. Our findings also concur with the study by Tonucci et al. [24], which demonstrated that phosphorylation of CIP4 by PKA promotes the formation of functional invadopodia and, thus, confirms PKA phosphorylation of *CIP4* as a regulator of the metastatic phenotype in cancer cells. Furthermore, the gene enrichment analysis results are in agreement with the Cheng et al. study [25], which demonstrated that PKA targets focal adhesion kinase for the promotion of cancer cell metastasis by cAMP. Moreover, Duan et al. [26] demonstrated that the phosphorylation of *TPI* at serine 58 by *PRKACA* enhances its enzymatic activity and glycolysis and, thus, its promotion of cancer growth and metastasis. Overall, our findings suggest that the primary function of *PRKACA* in cancer may be pro-migration/invasion rather than cell proliferation in GI cancers and may explain the seemingly paradoxical roles that have been described for *PRKACA* in different cancers [12]. For example, whilst PKA signaling has been demonstrated to regulate actomyosin contractility and cell migration, as well as membrane deformation, actin polymerization, cancer cell invasion, and metastasis, through phosphorylation of *CIP4* and *FAK* [23,24,25,26], activating *PRKACA* mutations have been associated with smaller tumor sizes of adrenocortical adenomas [27,28,29,30]. *PRKACA* mutations in these adenomas have been demonstrated to have a limited impact on cell proliferation and tumor growth [30].

The concept of a proliferation-invasion dichotomy in cancer cells is not new in oncology [31,32,33,34]. In many tumor types, individual cancer cells exist either as proliferating or invading cells within the tumor body, using various molecular mechanisms to switch between proliferative and invasive phenotypes [31,32,33,34]. Conversely, genes and genetic pathways that possess pro-metastasis, but not proliferation or even anti-proliferation, functions in tumorigenesis and tumor progression are not uncommonly activated in cancer. The *TGFB* pathway is a ready example of a pathway that has seemingly paradoxical roles in cancer. The *TGFB* pathway, an anti-cell proliferation pathway, is known to induce the epithelial-mesenchymal transition and cell migration in cancer cells [35]. Furthermore, the cell cycle regulator p16 inhibits tumor cell proliferation through impedance of the G1-S phase progression [36,37]. However, p16 has been demonstrated to be upregulated or overexpressed in the frontiers of invading tumor cell nests and in tumor buds, evidence that it is essential for tumor cell invasion and metastasis [38].

The most common deregulatory mechanisms of *PRKACA* expression include structural variation or gene fusion [13,14,15], somatic mutations [10,27,28,29,30], and copy number alterations [27,39]. Whilst no structural variation or somatic mutations of *PRKACA* were sought in this study, *PRKACA* copy number alterations were demonstrably a significant mechanism of *PRKACA* expression regulation in our gastrointestinal cancer cohorts. However, the predominant regulatory mechanisms that better predicted *PRKACA* expression in our gastric and colorectal cancer cohorts are epigenetic, transcriptional, and microRNA regulatory mechanisms. These aforementioned mechanisms are the most common gene regulatory mechanisms in cancers generally [40].

In conclusion, this study has demonstrated that a subset of GC and CRC exhibit elevated expression of *PRKACA,* which is associated with the overrepresentation of gene targets of multiple FDA-approved tyrosine kinase inhibitors and β-adrenergic signaling inhibitors. *PRKACA-*high tumors were also associated with adverse clinicopathological and molecular characteristics of GI cancers. Furthermore, the study showed that tumors with high *PRKACA* expression were enriched for biological and molecular functions associated with cell motility, cell invasion, and metastasis. The deregulatory mechanisms of epigenetics, miRNA, and copy number alteration mechanisms were found to predict *PRKACA* expression in GC and CRC.

## 4. Materials and Methods

### 4.1. Study Cohorts

This study retrospectively analyzed the clinicopathological and genomic data of the cancer genome atlas (TCGA) colon, rectal, and gastric cancer cohorts [41,42,43,44]. All the data were retrieved from the Genome Data Commons (GDC) and cBioPortal for Cancer Genomics databases. Transcript quantification was accomplished with RNASeq (mRNA) and miRNASeq (miRNA). Methylation beta values were generated using a methylation array on the Illumina Human Methylation 450 platform, while the masked copy number segment data were generated using the Affymetrix SNP 6.0 genotyping array.

### 4.2. Data Processing

The clinico-genomic data of the GI cancer cohorts were retrieved from the GDC and CBioPortal databases using Linux-based scripts and codes that were written in the Windows-based Ubuntu 20.04 environment. Linux-based codes and scripts were also utilized to prepare the gene expression datasets in accordance with the Molecular Signature Database (MSigDB) [45,46] and DESeq2 Gene Enrichment Analyses requirements (https://cloud.genepattern.org/, accessed on 25 July 2024) [47], whereas Excel spreadsheet was utilized in the generation of the phenotype and derivative gene set files (see below), following which these were converted to cls and grp files, respectively. The gastric cancer cohort comprised 441 primary cases with clinicopathological (including prognostic data), RNASeq, chromosomal copy number segment, methylation, and somatic mutation data. The following data were available for this cohort: clinicopathological (between 380 and 441 of 441 cases for each clinicopathological indices; Table 1); mRNA expression (415/441 cases); chromosomal copy number segment (441/441 cases); methylation (between 393 and 440 of 441 cases for individual methylation loci); microRNA expression (441/441 cases) data.

The colorectal cancer cohort included 629 primary cases with the following amount of data: clinicopathological (between 545 and 629 of 629 cases for each clinicopathological indices; Table 1); mRNA expression (534/629 cases); chromosomal copy number segment (512/629 cases); methylation (between 331 and 524 of 629 cases for individual methylation loci); microRNA expression (506/629 cases) data.

The *PRKACA* expression data from these cohorts were converted into normally distributed data by using the method described by Templeton [48] before their utilization for statistical analyses.

### 4.3. Study Approach

First, GSEA and drug-target annotation with Drug Signature Database (DSigDB) gene set libraries in the Enrichr environment (https://maayanlab.cloud/Enrichr/enrich, accessed on 25 July 2024) [45,46,49,50,51] were utilized to infer the potential for *PRKACA* expression to predict susceptibility or resistance to drug agents. Then, the relationship between the clinicopathological indices and *PRKACA* expression in each of the GI cancers was sought in the cohorts using the appropriate statistical tests. In addition, the biological significance of *PRKACA* expression in GI carcinogenesis was confirmed using GSEA with the GC and CRC gene expression datasets and the MSigDB Hallmark gene sets [45,46]. The list of genes in the core enrichment of the significant gene sets was applied to gene ontology analysis in the Enrichr environment [45,46,49,50] to verify the biological, molecular, and functional significance of the enriched genes. Moreover, the mechanisms of altered *PRKACA* expression were sought in each of the GI cancer cohorts using the copy number segment, structural variation, methylation (beta values), transcription factor, and miRNA expression data. *PRKACA*-targeting transcription factors were retrieved from the transcription factor database TF2DNA_DB (https://www.fiserlab.org/tf2dna_db/search_genes.html, accessed on 25 July 2024) [52], and their expression values were correlated with *PRKACA* expression in either cancer cohorts using bivariate analyses. Significantly correlated transcription factors were then selected for regression analyses. Differential miRNA enrichment was sought between *PRKACA*-low and *PRKACA*-high cases using the online DESeq2 software v.3 on the GenePattern computing environment [47]. Gene enrichment analyses by DESeq2 were used to identify the top 40 miRNA species that differentially expressed in the *PRKACA*-low and -high subsets in the GC and CRC cohorts. Significantly enriched miRNAs, which also showed correlations with *PRKACA* expression by bivariate analyses, were then incorporated into a regression analysis, together with the methylation, *PRKACA*-targeting transcription factors, structural variation, and *PRKACA* copy number indices, to infer their roles in the deregulation of *PRKACA* expression.

### 4.4. Statistical Analyses

The enrichment analysis in the GSEA and DESeq2 software was performed using the software’s default parameters. GSEA was performed as a phenotype permutation. Gene ontology enrichment and drug-target association annotations were performed in the Enrichr environment using thresholds nominal *p*-value ≤ 0.05 and FDR ≤ 0.05. The clinicopathological and genomic data of the cancer cohorts were input into SPSS version 29. The Chi-square (or Fisher) test was used to probe for significant associations between categorical variables, while bivariate correlative analysis was utilized to test the correlations between continuous variables. Multivariate analysis of *PRKACA* expression correlation was investigated with multiple linear regression and binary logistic regression analyses. The one-way ANOVA test was used to measure the mean differences of continuous variables between discrete groups. The prognostic significance of *PRKACA* expression was defined using Kaplan–Meier and Cox regression analyses. A *p*-value of <0.05 was used as the threshold for significant tests, while the Benjamini–Hochberg correction was used to correct for multiple testing at an FDR of <0.05.

## Figures and Tables

**Figure 1 pharmaceuticals-17-01263-f001:**
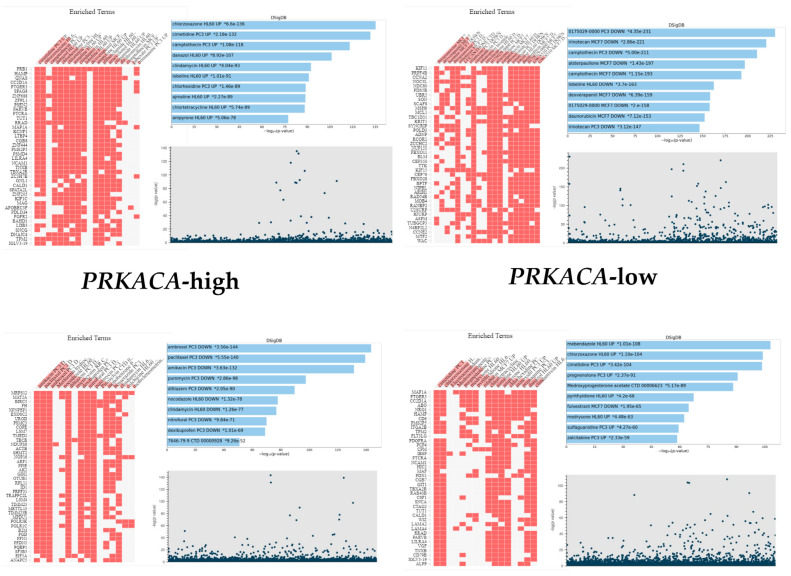
Clustergram, bar chart, and Manhattan plot showing enrichment of gene targets of multiple therapeutic agents in the two *PRKACA* subsets of GC (Upper panel) and CRC (Lower panel). The full names of the ontology terms are listed in Appendix A Drug-targets_PRKACA_high_GC and Drug-targets_PRKACA_low_GC.

**Figure 2 pharmaceuticals-17-01263-f002:**
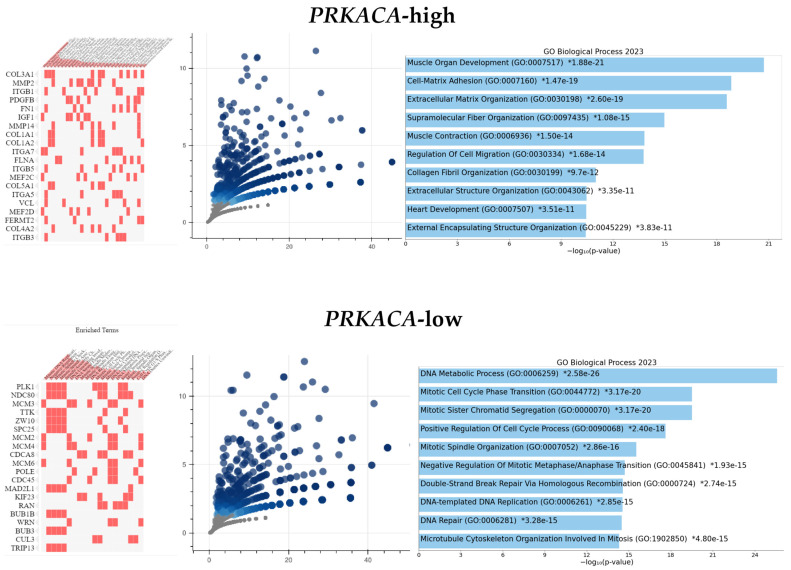
Gene ontology enrichment in the *PRKACA* subsets of GC. The upper panel displays cluster gram, volcano plot, and a bar chart showing the enrichment of biological functions involved in cell migration/invasion in the *PRKACA*-high GC cohort. The lower panel shows charts that display the enrichment of biological processes associated with cell proliferation in the *PRKACA*-low GC cohort. Details of the gene ontology terms can be found in the Appendix A GC_GSEA.PRKACA_high and GC_GSEA.PRKACA_low.

**Figure 3 pharmaceuticals-17-01263-f003:**
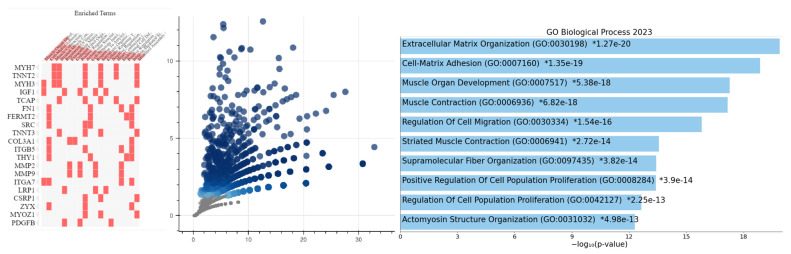
Gene Ontology Enrichment Analysis of PRKACA-high subset of the CRC cohort. Clustergram, volcano plot, and bar chart showing enrichment of biological processes involved in cell proliferation. Details of the ontology terms and gene names displayed on the images are available in the Appendix A CRC_GSEA.PRKACA_high.

**Figure 4 pharmaceuticals-17-01263-f004:**
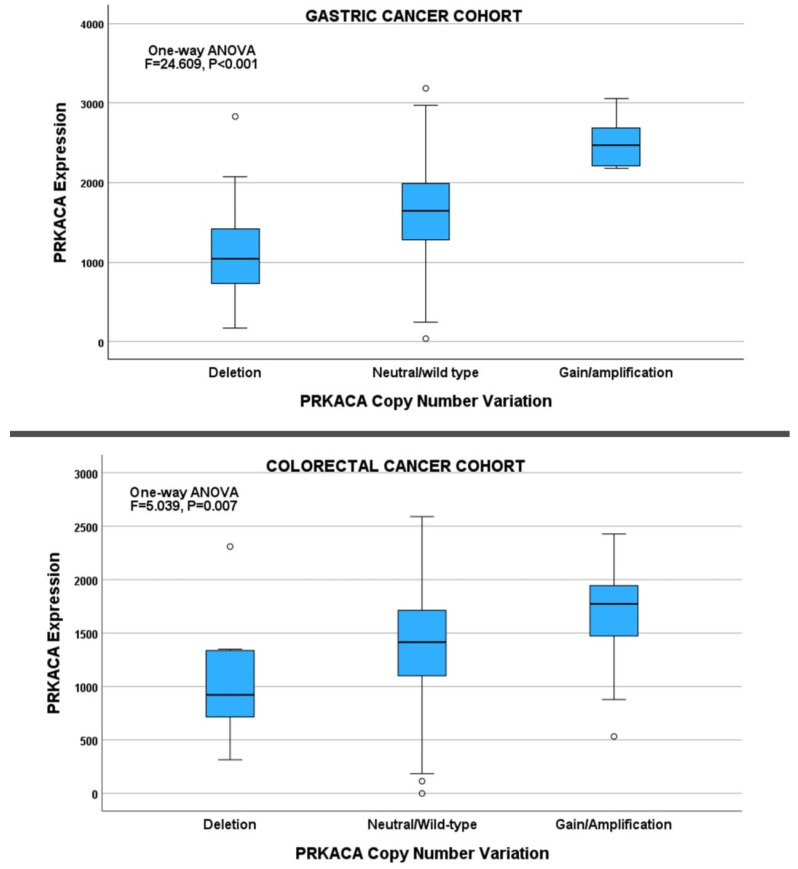
Box plots showing significant correlations between *PRKACA* expression and copy number alterations in the GC (**Upper**) and CRC (**Lower**) cohorts.

**Figure 5 pharmaceuticals-17-01263-f005:**
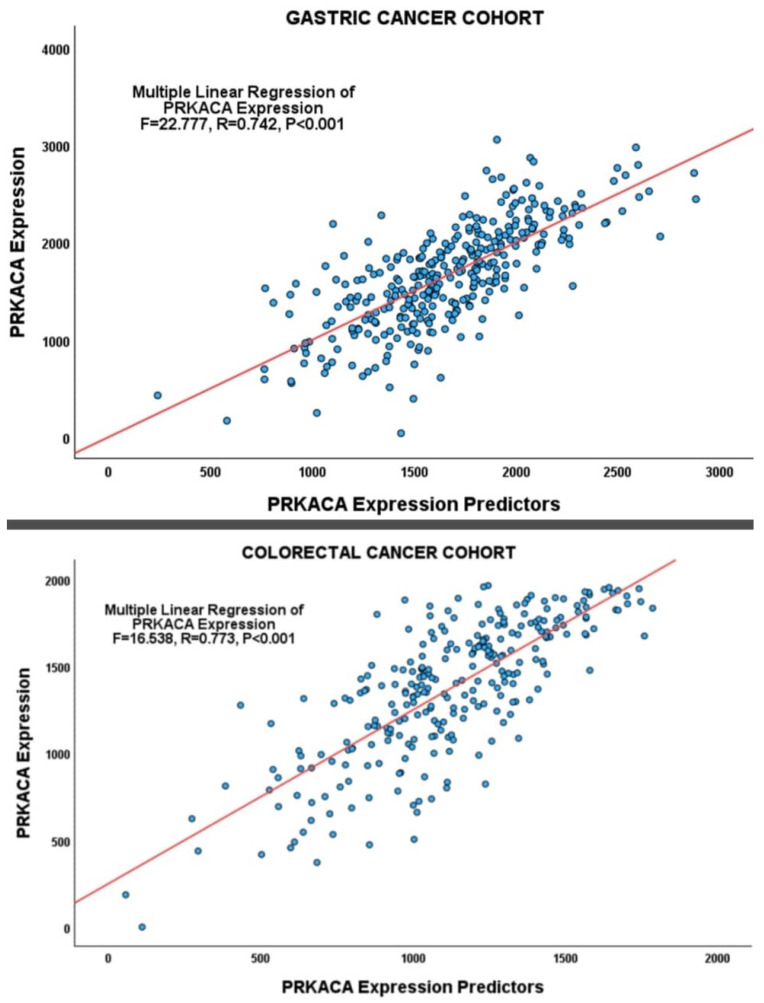
Scatterplots showing significant relationships between *PRKACA* expression and the regressors (*PRKACA* methylation, *PRKACA* copy number alterations, *PRKACA*-targeting transcription factors, and miRNA species) in the GC (**Upper**) and CRC (**Lower**) cohorts.

**Table 1 pharmaceuticals-17-01263-t001:** Clinicopathological correlates of *PRKACA* expression in gastric and colorectal cancers.

*PRKACA* Expression							
Clinicopathological Features		Low *PRKACA*	High *PRKACA*	Total	X^2^ *	*p* Value	Adjusted *p* Value
Gastric cancer cohort							
Age Group	30–60 yrs	52	79	131	7.630	0.006	0.016
	61–90 yrs	150	126	276			
	Total	202	205	407			
Gender	Male	137	128	265	0.857	0.355	0.441
	Female	69	78	147			
	Total	206	206	412			
Race/Ethnicity	White	126	133	259	2.609	0.112	0.179
	Native Americans	1	0	1			
	Black	10	2	12			
	Asians	31	56	87			
	Total	168	191	359			
Pathological tumor stage	pT1	13	9	22	0.753	0.386	0.441
	pT2	38	49	87			
	pT3	85	96	181			
	pT4	63	51	114			
	Total	199	205	404			
Pathological nodal stage	N0	46	77	123	9.775	0.002	0.008
	N1	54	55	109			
	N2	48	31	79			
	N3	46	36	82			
	Total	194	199	393			
Pathological metastasis stage	M0	183	181	364	0.280	0.596	0.596
	M1 and above	15	12	27			
	Total	198	193	391			
TNM stage	Early Stage	45	81	126	13.226	<0.001	0.002
	Late Stage	152	123	275			
	Total	197	204	401			
Histological type of gastric cancer	Diffuse type Adenocarcinoma	10	3	13	3.892	0.049	0.098
	Intestinal type Adenocarcinoma	196	203	399			
	Total	206	206	412			
Colorectal cancer cohort							
Age Group	31–60 yrs	70	103	173	9.295	0.002	0.007
	61–90 yrs	198	165	363			
	Total	268	268	536			
Gender	Male	141	142	283	0.007	0.931	0.931
	Female	127	126	253			
	Total	268	268	536			
Race/Ethnicity	White	116	167	283	3.876	0.049	0.106
	Asian/Native American	7	6	13			
	Black	34	29	63			
	Total	157	202	359			
Pathological tumor stage	pTis and pT1	5	10	15	0.120	0.729	0.790
	pT2	55	33	88			
	pT3	172	196	368			
	pT4	36	29	65			
	Total	268	268	536			
Pathological nodal stage	N0	170	134	304	7.160	0.007	0.018
	N1	55	76	131			
	N2	43	56	99			
	Total	268	266	534			
Pathological metastasis stage	M0	202	192	394	0.272	0.602	0.711
	M1	35	38	73			
	Total	237	230	467			
TNM stage	Early Stage	153	129	282	3.956	0.0467	0.106
	Late Stage	115	137	252			
	Total	268	266	534			
Histological type of colorectal cancer	Adenocarcinoma NOS	225	240	465	3.452	0.063	0.117
	Mucinous Adenocarcinoma	40	26	66			
	Total	265	266	531			
Colonic tumor site 1	Right colon	164	92	256	23.553	<0.001	<0.001
	Left colon	92	128	220			
	Total	256	220	476			
Primary tumor site	Colon	268	177	445	108.404	<0.001	<0.001
	Rectum	0	90	90			
	Total	268	267	535			
Vascular Invasion (VI)	VI absent	180	174	354	1.746	0.186	0.269
	VI present	48	62	110			
	Total	228	236	464			
Lymphovascular Invasion	LVI absent	153	146	299	0.513	0.474	0.616
	LVI present	87	95	182			
	Total	240	241	481			
Perineural Invasion (PNI)	PNI absent	75	95	170	1.884	0.170	0.269
	PNI present	20	39	59			
	Total	95	134	229			

* Pearson Chi-square test for 2 × 2 tables, and Linear-by-linear association test for >2 × 2 tables; TNM = Tumor, Node and Metastasis.

**Table 2 pharmaceuticals-17-01263-t002:** Molecular correlates of *PRKACA* expression in gastric and colorectal cancers.

*PRKACA* Expression							
Molecular Correlates		Low *PRKACA*	High *PRKACA*	Total	X^2^ *	*p* Value	Adjusted *p* Value
Gastric cancer cohort							
Molecular subtypes	CIN	120	102	222	7.231	0.007	0.014
	MSI	42	31	73			
	GS	18	32	50			
	EBV	11	19	30			
	POLE	2	5	7			
	Total	193	189	382			
MSS vs. MSI	MSS	140	138	278	1.192	0.275	0.275
	MSI	42	31	73			
	Total	182	169	351			
Aneuploidy score	Low Aneuploidy Score	88	110	198	5.563	0.018	0.024
	High Aneuploidy Score	112	87	199			
	Total	200	197	397			
Global methylation score	Low Global methylation score	128	74	202	28.455	<0.001	<0.001
	High Global methylation score	77	131	208			
	Total	205	205	410			
Colorectal cancer cohort							
Molecular subtypes	CIN	112	178	290	21.901	<0.001	<0.001
	MSI	42	21	63			
	GS	39	16	55			
	POLE	3	4	7			
	Total	196	219	415			
MSS vs. MSI	MSS	154	198	352	11.260	0.001	0.002
	MSI	42	21	63			
	Total	196	219	415			
Mutation Count	Low Mutation count	93	120	213	6.168	0.013	0.017
	High Mutation count	136	110	246			
	Total	229	230	459			
Fraction of Genome Altered	Low Fraction Genome Altered	151	112	263	10.592	0.001	0.002
	High Fraction Genome Altered	102	136	238			
	Total	253	248	501			
MANTIS Score	Low MANTIS Score	109	116	225	0.724	0.395	0.395
	High MANTIS Score	135	123	258			
	Total	244	239	483			
MSI Sensor	Low MSI Sensor Score	106	151	257	19.578	<0.001	<0.001
	High MSI Sensor Score	150	96	246			
	Total	256	247	503			
Aneuploidy Score	Low Aneuploidy Score	120	89	209	8.537	0.003	0.005
	High Aneuploidy Score	92	121	213			
	Total	212	210	422			
Tumor Mutational Burden	Low TMB	83	108	191	3.701	0.054	0.061
	High TMB	115	102	217			
	Total	198	210	408			

* Pearson Chi-square test for 2 × 2 tables, and Linear-by-linear association test for >2 × 2 tables; CIN = chromosomal instability, MSI = microsatellite instability, EBV = Epstein-Barr Virus, GS = genome stable, POLE = polymerase e, MSS = microsatellite stable, MANTIS = Microsatellite Analysis for Normal Tumor InStability (computational tool for assessing the extent of MSI in cancers), MSI Sensor = an alternative computational tool for assessing MSI in cancers.

**Table 3 pharmaceuticals-17-01263-t003:** Deregulation of *PRKACA* expression in gastric cancer *.

Gastric Cancer				
R	R^2^		Adjusted R^2^	S.E. of Estimate
0.742	0.550		0.526	366.808
Coefficients				
	Unstandardized Coefficients		t	*p*
	B	S. E.		
(Constant)	2453.432	240.529	10.200	<0.001
hsa-mir-490	0.586	0.130	4.514	<0.001
*PRKACA* CNV	450.825	69.936	6.446	<0.001
*ZNF644* Expression	−60.541	15.418	−3.927	<0.001
*ZNF557* Expression	79.896	24.840	3.216	0.001
cg17818798	3007.832	745.072	4.037	<0.001
cg26613742	−2233.574	373.632	−5.978	<0.001
cg01596520	1031.285	338.217	3.049	0.002
*HEY2* Expression	71.020	18.869	3.764	<0.001
*CDC5L* Expression	−8.872	4.218	−2.103	0.036
*TFEB* Expression	21.175	7.597	2.787	0.006
cg19621460	447.432	160.599	2.786	0.006
hsa-mir-7641-1	13.596	4.805	2.830	0.005
*SP8* Expression	−61.200	21.970	−2.786	0.006
*ZNF101* Expression	45.030	16.502	2.729	0.007
*ZKSCAN3* Expression	128.247	46.163	2.778	0.006
*ZNF451* Expression	−97.305	34.028	−2.860	0.005
cg01010868	−29,345.397	12879.446	−2.278	0.023
ANOVA				
	df		F	*p*
Regression	17		22.777	<0.001
Residual	317			
Total	334			

* Multiple linear regression analysis.

**Table 4 pharmaceuticals-17-01263-t004:** Deregulation of *PRKACA* expression in colorectal cancer *.

R	R^2^			Adjusted R^2^	S.E. of Estimate
0.773	0.598			0.561	259.799
Coefficients					
		Unstandardized Coefficients		t	*p*
		B	S. E.		
(Constant)		−520.078	360.444	−1.443	0.150
*ZBTB7A* Expression		31.065	4.177	7.437	<0.001
hsa-mir-143		0.001	0.000	2.461	0.015
cg15814923		−6980.818	2832.849	−2.464	0.014
hsa-mir-577		−0.505	0.196	−2.578	0.011
*TSHZ3* Expression		92.833	19.104	4.859	<0.001
*DHX34* Expression		18.602	6.649	2.798	0.006
cg17818798		−996.038	497.031	−2.004	0.046
cg19586199		−610.229	186.442	−3.273	0.001
*ZNF451* Expression		−124.101	27.985	−4.434	<0.001
*ZNF442* Expression		324.096	163.862	1.978	0.049
*ZKSCAN8* Expression		68.663	15.481	4.435	<0.001
*MAFK* Expression		−15.686	4.159	−3.772	<0.001
*HOXA1* Expression		80.336	23.028	3.489	0.001
*ZNF516* Expression		−111.062	23.760	−4.674	<0.001
cg17119568		2626.810	942.395	2.787	0.006
hsa-mir-216a		20.726	7.769	2.668	0.008
*ZNF140* Expression		−37.664	20.633	−1.825	0.069
*ZNF557* Expression		89.689	34.613	2.591	0.010
hsa-mir-1-2		−1.278	0.643	−1.986	0.048
cg20110535		846.904	368.866	2.296	0.023
hsa-mir-5092		−78.546	31.141	−2.522	0.012
*ZNF235* Expression		−162.614	78.187	−2.080	0.039
ANOVA					
	df			F	*p*
Regression	22			16.538	<0.001
Residual	245				
Total	267				

* Multiple linear regression analysis.

## Data Availability

All the genomic and clinicopathological data utilized for this study are freely available in the cBioPortal for Cancer Genomics website (https://www.cbioportal.org/, accessed on 25 June 2024), and the Genome Data Commons repository (https://portal.gdc.cancer.gov/analysis_page, accessed on 25 June 2024).

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
