# Peer review of "Therapeutic, Clinicopathological, and Molecular Correlates of PRKACA Expression in Gastrointestinal Cancers"

_pharmaceuticals, 2024, doi:10.3390/ph17101263_

Round 1

Reviewer 1 Report

Comments and Suggestions for Authors

In the manuscript titled “Therapeutic, clinicopathological, and molecular correlates of PRKACA expression in gastrointestinal cancers,” the authors examined the clinicopathological characteristics of gastric and colorectal cancer patients classified as PRKACA-low and PRKACA-high. They also investigated the correlation between PRKACA expression and miRNAs and transcription factors. While the topic is intriguing, several issues need addressing:

1. The authors should clarify their choice of FDR < 0.25 for gene ontology enrichment analysis instead of more stringent cutoffs like FDR < 0.1 or FDR < 0.025.

2. Detailed criteria for defining PRKACA-low and PRKACA-high cases should be explained. What method was used for categorization, and how many cases were in each group?

3. All abbreviations in Tables 1 and 2 should be defined at the bottom of each table.

4. The methods section should specify how PRKACA-targeting transcription factors were identified and the criteria applied.

5. Some study results, particularly concerning miRNAs and transcription factors, were not discussed in the results section.

6. The presentation of gene ontology terms is overly dense, making it hard to read. It would be better to highlight three significant terms and display others in figures.

Author Response

Reviewer’s comment: In the manuscript titled “Therapeutic, clinicopathological, and molecular correlates of PRKACA expression in gastrointestinal cancers,” the authors examined the clinicopathological characteristics of gastric and colorectal cancer patients classified as PRKACA-low and PRKACA-high. They also investigated the correlation between PRKACA expression and miRNAs and transcription factors. While the topic is intriguing, several issues need addressing:

Authors’ response: Thank you for reviewing our manuscript.

Reviewer’s comment 1: The authors should clarify their choice of FDR < 0.25 for gene ontology enrichment analysis instead of more stringent cutoffs like FDR < 0.1 or FDR < 0.025.

Authors’ response 1: The FDR threshold for gene ontology enrichment by the Enrichr tool that was used in this study is <0.05. We have corrected the error in the manuscript.

Reviewer’s comment 2: Detailed criteria for defining PRKACA-low and PRKACA-high cases should be explained. What method was used for categorization, and how many cases were in each group?

Authors’ response 2: “Normalised PRKACA expression values were dichotomised using the mean PRKACA values for each cohort. There were 206 PRKACA-low (50.0%) and 206 PRKACA-high (50.0%) of GC cases”. Similarly, for the CRC cohort, “normalised PRKACA expression were dichotomised into PRKACA-low (N=268, 50%), and PRKACA-high subsets (N=268, 50%).” These statements have been added to the appropriate sections of the manuscript.

Reviewer’s comment 3: All abbreviations in Tables 1 and 2 should be defined at the bottom of each table.

Authors’ response 3: All abbreviations in Tables 1 and 2 have been defined at the bottom of each table. “TNM = Tumour, Node and Metastasis” (Table 1)

“CIN=chromosomal instability, MSI=microsatellite instability, EBV=Epstein-Barr Virus, GS=genome stable, POLE=polymerase e, MSS=microsatellite ‎stable, MANTIS= Microsatellite Analysis for Normal Tumor InStability (computational tool for assessing the extent of MSI in cancers), MSI Sensor= an ‎alternative computational tool for assessing MSI in cancers” (Table 2). ‎Added to the manuscript

Reviewer’s comment 4: The methods section should specify how PRKACA-targeting transcription factors were identified and the criteria applied.

Authors’ response 4: “PRKACA-targeting ‎transcription factors were retrieved from the transcription factor ‎database, TF2DNA_DB (https://www.fiserlab.org/tf2dna_db/search_genes.html) [56], ‎and their expression values were correlated with PRKACA expression in either cancer ‎cohorts using bivariate analyses. Significantly correlated transcription factors were then ‎selected for regression analyses.”‎ Added to the manuscript

Reviewer’s comment 5: Some study results, particularly concerning miRNAs and transcription factors, were not discussed in the results section.

Authors’ response 5: We discussed the details of miRNAs and transcription factors contributions to altered PRKACA expression in either cohort and displayed the regression analyses results in Table 3 and Table 4. Furthermore, we noted in the Discussion section that miRNAs and transcriptional regulations are common mechanisms, among others, of gene regulation in cancers.

Reviewer’s comment 6: The presentation of gene ontology terms is overly dense, making it hard to read. It would be better to highlight three significant terms and display others in figures.

Authors’ response 6: A few, most relevant gene ontologies have been selected and left in the Result sections, leaving other significant ontologies in the Figures

Reviewer 2 Report

Comments and Suggestions for Authors

Attached.

Author Response

Comments Manuscript ID: pharmaceuticals-3174442

Manuscript type: Research paper

Journal: Pharmaceuticals

Reviewer’s comment: The study is a retrospective analysis where the authors are trying to correlate the expression levels of PRKACA to clinical outcomes and patient prognosis. It is a well-performed statistical analysis in good detail. The conclusions are easily linked to the results. Additionally, it is within the scope of Pharmaceuticals.

Authors’ response: Thank you for your excellent review of our manuscript

Reviewer’s comment 1: In the abstract, authors must put at least one sentence explaining the importance of PRKACA and why they are studying it on GC and CRC.

Authors’ response 1: The statement “PRKACA alterations have clear diagnostic and biological roles in the fibrolamellar variant of hepatocellular ‎carcinoma, and a potential predictive role in that cancer type” has been added to the Abstract section. ‎

Reviewer’s comment 2: Be careful with the acronyms. Once an abbreviation is introduced, it must be used throughout the text.

Authors’ response 2: We thank the reviewer for this observation.

Reviewer’s comment 3: The sections are not numbered correctly. For example, 2. Results and then 3.1

Authors’ response 3: We have now corrected this oversight in the manuscript

Reviewer’s comment 4: The title of the sections is sometimes too long. They must be informative but short.

Authors’ response 4: We have now shortened Result section 2.1. to read “Enrichment of gene targets of multiple drug agents in PRKACA-low and -high cancers”, instead of “Enrichment of gene targets of tyrosine kinases, chemotherapeutics and ‎β-adrenergic inhibitors in PRKACA-low and -high cancers”

Reviewer’s comment 5: Since the authors use numerous statistical methods, a brief description of each method's outcome would be a plus.

Authors’ response 5: We used standard statistical tests along with computational methods such as gene enrichment and gene ontology enrichment analyses. The results from these statistical analyses were clearly stated in the Result sections, and displayed as tables, figures, graphs and charts. But we will be grateful to have further clarifications about what the reviewer means, with a few specific examples, by “a brief description of the method’s outcome”, if this response does not meet up with the reviewer’s expectations.

Reviewer’s comment 6: It would be nice to have some examples when referring to a gene (line 457).

Authors’ response 6: Detailed lists of genes in the enriched gene ontologies are found in the Supplementary Materials: GOEA_GC_PRKACA_low, GOEA_GC_PRKACA_high, and GOEA_CRC_PRKACA_high. We suppose that the reviewer received the Supplementary Materials for the review purposes.

Reviewer’s comment 7: Some figures are difficult to read due to their small letter size.

Authors’ response 7: The figures were produced with the expectation that when the manuscript is published online, the reader will be able to enlarge the figures and have a detailed view of them. But if our assumption is wrong, we will defer to the editors and re-produce magnified figures.

Round 2

Reviewer 1 Report

Comments and Suggestions for Authors

The authors have thoroughly addressed all comments, and the manuscript now appears to be in excellent shape.